# The Impact of Mutant *EDNRB* on the Two-End Black Coat Color Phenotype in Chinese Local Pigs

**DOI:** 10.3390/ani15040478

**Published:** 2025-02-07

**Authors:** Min Huang, Zuohao Wen, Tao Huang, Xiaolong Zhou, Zhijun Wang, Songbai Yang, Ayong Zhao

**Affiliations:** College of Animal Science and Technology · College of Veterinary Medicine, Zhejiang A&F University, Hangzhou 311300, China; minhuang0702@outlook.com (M.H.); wen824345107@outlook.com (Z.W.); taohuang@zafu.edu.cn (T.H.); zhouxiaolong@zafu.edu.cn (X.Z.); zhijunwang@zafu.edu.cn (Z.W.); sbyang@zafu.edu.cn (S.Y.)

**Keywords:** Chinese two-end black coat color pigs, *EDNRB*, melanin pathway, cell migration, protein interaction

## Abstract

*Endothelin Receptor Type B* is a gene expressed in various cells during the embryonic stage, including melanocyte precursor cells. An 11 bp deletion in Endothelin Receptor Type B was previously found that linked to the two-end black coat color in Chinese pigs. This study aimed to investigate how this mutation affects two-end black coat color formation. We constructed plasmids for both wild and mutant *Endothelin Receptor Type B*, as well as *Endothelin-1*, and transfected them into mouse melanoma cells. Results showed that the mutation significantly reduced the expression of key melanin pathway genes (*phospholipase Cγ*, *rapidly accelerated fibrosarcoma*, and *microphthalmia-associated transcription factor*), melanin content, and cell migration ability. Additionally, the mutant Endothelin Receptor Type B could not interact with the Endothelin-1 protein. These findings suggest that the 11 bp deletion disrupts melanin production and melanocyte function, offering insights into the molecular mechanism underlying two-end black coat color in Chinese pigs. This study enhances our understanding of pigmentation genetics and its potential applications in animal breeding and genetic studies.

## 1. Introduction

EDNRB is a G protein-coupled receptor with seven transmembrane regions; it is expressed in a variety of cells, including melanocyte precursors and intestinal neuronal precursor cells, during the embryonic stage [1,2]. Previous studies have shown that when the *EDNRB* gene is mutated or knocked out, the related melanin signaling pathway will be disrupted in mice, which leads to the abnormal development of melanocyte precursors and appears almost entirely white except for a small amount of black coat on the head and buttocks [3,4,5]. In pigmentation-related studies, there is a kind of endothelin (EDN) that is the ligand of EDNRB, which can promote differentiation and affect the migration of immature melanocytes, and mainly stimulates the proliferation and division of mature melanocytes [6,7]. Studies have shown that the melanin pathway can affect the migration of melanocyte precursors. At 11.5 days of embryonic age in mice, obstruction of the EDNRB signaling pathway can cause abnormal development of melanocytes in mouse skin and affect the proliferation of melanocyte precursors [5,8]. It can be seen that the EDNRB signaling pathway plays a certain role in regulating the proliferation of melanocytes [9,10].

Coat color is determined by the distribution of melanocytes and melanin types [11]. Studies have revealed the mechanism of melanoblast migration and differentiation in early embryonic development for the pigs’ coat color, which explains the origin and domestication process of pigs to a certain extent [12,13]. Chinese indigenous pig breeds have a rich variety of coat color types, including black pigs, white pigs, brown pigs, large black–white pigs, and other coat color phenotypes. Among them, the two-end black (TEB) coat color phenotype is characterized by a black head and tail, and white body (Figure 1A) [14]. The results show that *EDNRB* is a causal gene affecting the coat color phenotype of TEB pigs in China [15,16]. Previous studies inferred from SNP analysis that *EDNRB* mutations are responsible for the TEB phenotype [17]. One 25-SNPs region of *EDNRB* is a strong candidate for pathogenic mutation, which can regulate the TEB coat color in Jinhua pigs by altering enhancer function [18]. In order to reveal the evolutionary selection mechanism of coat color formation in TEB pigs, it was found that the *EDNRB* gene had 11 bp deletion on chromosome 11: 50,076,945–50,076,960 bp, which led to the premature occurrence of stop codons in the *EDNRB* transcript and resulted in the truncated amino acid (aa) polypeptide of the EDNRB protein (Figure 1B) [19]. However, how the 11 bp mutation affects melanin formation and, consequently, the TEB phenotype in Chinese pigs is not yet known. In order to further explore the effect of the mutant *EDNRB* (harboring the 11 bp deletion) on the coat color phenotype of Chinese TEB pigs, we conducted experiments based on the mutant-type *EDNRB* to verify the effect of the mutation on the coat color formation of Chinese TEB pigs.

In this study, we first synthesized sequences of wild-type *EDNRB*, mutant-type *EDNRB*, and *EDN1*, and then constructed recombinant plasmids for them. Subsequently, we transfected these recombinant plasmids into mouse melanoma cells (B16), and detected the expression levels of *PLCγ*, *Raf*, and *MITF* genes in the melanin pathway by real-time fluorescent quantitative PCR (RT-qPCR). The results showed that the mutant-type *EDNRB* significantly reduced the expression levels of the three genes. The detection of melanin expression assay showed that mutant-type *EDNRB* significantly reduced melanin production more than that of the wild type. In addition, wound-healing experiments confirmed that mutant-type *EDNRB* significantly reduced the migration rate of melanocytes, and the protein produced could not interact with the EDN1 protein. In summary, this study confirmed that the mutant-type *EDNRB* with the 11 bp deletion could not bind to EDN1 due to the truncated amino acid polypeptide of EDNRB, which affected the expression of related genes in the melanin pathway and the production of melanin. In addition, the mutant-type *EDNRB* decreased the migration rate of melanocytes at the same time, which finally resulted in the TEB coat color of pigs.

## 2. Materials and Methods

### 2.1. Plasmid Construction

First of all, we looked through the Ensembl website and obtained the cds sequences of *EDNRB* and *EDN1* of pigs. The cds sequences of mutant EDNRB came from the previous study (Appendix A) [19]. The above three target sequences were constructed onto overexpressed vectors containing Flag and HA tags, and fused with labeled antibodies for expression. That is, the recombinant plasmids of wild-type *EDNRB* (p3xFLAG-CMV-14-EDNRB), mutant-type *EDNRB* (p3xFLAG-CMV-14-EDNRB_M), and *EDN1* (pCMV-N-HA-EDN1) were constructed. Plasmid construction was performed by Tsingke Biological Company (Beijing, China). Subsequently, TOYOBO’s (Novi, MI, USA) kod One™ PCR Master Mix-Blue item number KMM-201 was used. According to the reference system provided in the instruction manual, the bacterial solution, the universal primers provided by Tsingke, deionized water, and PCR enzyme were mixed, and after preparing the mixed solution, PCR was performed on the bacterial solution to verify whether the sequence was successfully constructed. At the same time, the bacterial solution was shaken and expanded, and the cultured bacterial solution was sent to Tsingke Biological Company (Beijing, China) for sequencing. PCR amplification was carried out as follows: denaturation of 98 °C for 10 s, followed by 35 cycles of 60 °C for 40 s, and a specific extension temperature of 60 °C for 10 s.

### 2.2. Detection of Melanin Pathway Gene Expression

The B16 cells were inoculated into 6-well plates. After cell division and proliferation to 70–80%, the overexpression vector was transfected into B16 cells according to the transfection instructions of Lipofectamine™ 3000 liposome from Invitrogen CA (Waltham, MA, USA). The cells were then cultured in an incubator at 37 °C and 5% CO_2_ and cultured for 24 h using the prepared 1640 complete medium. During this period, the color change of the culture medium was observed. If the color turned yellow, new culture medium could be considered. Total RNA for B16 cells was extracted by Trizol method, and cDNA was synthesized by a 5× All-In-One RT MasterMix transcription kit (ABM, Calgary, AB, Canada). Then SnapGene 6.0.2 software was used to design primers for genes, including *PLCγ*, *Raf*, *MITF,* and the internal reference gene GAPDH. The primers were provided by Qingke Biotechnology Company Limited (Beijing, China) (Appendix A). The cDNA template synthesized by reverse transcription was added to the verified primers, and a 10 μL mixed solution was prepared using Nova Bio (Shanghai, China)’s 2xS6 Universal SYBR qPCR Mix. The mixture was used for RT-qPCR to detect the effects of wild-type *EDNRB* and mutant *EDNRB* on the melanin pathway. RT-qPCR amplification was carried out as follows: Initial denaturation of 95 °C for 20 s, followed by 40 cycles of 95 °C for 3 s, and a specific annealing temperature of 60 °C for 30 s.

### 2.3. Detection for Cell Melanin Expression

Separately, the three plasmids p3xFLAG-CMV-14-EDNRB, p3xFLAG-CMV-14-EDNRB_M, and pCMV-N-HA-EDN1 were divided into pairs, and plasmids p3xFLAG-CMV-14-EDNRB and pCMV-N-HA-EDN1 were co-transfected into B16 cells using Lipofectamine 3000, following the manufacturer’s protocol. The cells were washed with PBS to remove the residual medium, and we added 200 μL of Beyotime RIPA Lysis Buffer (Strong) (Beijing, China) to each well. Among them, RIPA Lysis Buffer needs to be added to Beyotime’s Protease inhibitor cocktail for general use at a ratio of 1:50. The plates were placed on ice and incubated for 15 min. The cells were scraped and the extracts were separated by a cooled microcentrifuge at a centrifugal force of 14,000 rpm for 10 min, and then we removed the supernatant. The mouse melanin ELISA kit, provided by Basediao Biotechnology Company Limited (Beijing, China), was used to detect the cellular melanin content. According to the instructions, we first diluted the standard provided in the kit as required, then we set up blank wells (no sample and enzyme-labeled reagent, the rest of the steps are the same), standard wells, and sample wells to be tested, added 50 μL of sample to be tested to the standard wells, added 40 μL of sample diluent to the sample wells to be tested, and then added 10 μL of sample to be tested (the final dilution of the sample is 5 times). Then we placed it into a 37 °C incubator for incubation, followed by 5 washes with the washing solution in the kit, and then we added 50 μL of enzyme-labeled reagent to each well (except the blank well), repeated the incubation and washing again, and then we used the color developer to develop at 37 °C in the dark for 10 min, an added 50 μL of stop solution to terminate the reaction (the color turns from blue to yellow at this time). After obtaining the final product, we adjusted the blank well to zero, then the absorbance (OD value) of each hole was measured at 450 nm wavelength in sequence, and the linear regression equation of the standard curve was calculated with the concentration and OD value of the standard substance. The OD value of the sample was substituted into the equation, the sample concentration was calculated, and then we multiplied by the dilution ratio, which is the actual melanin concentration of the sample.

### 2.4. Measurement of Cell Migration

After culturing cells for 24 h in a 6-well plate, a scratch was created using a 10 μL pipette tip. Cells were washed twice with PBS to remove debris, and fresh Dulbecco’s Modified Eagle Medium (DMEM) without FBS was added. An inverted biological microscope was used for observation at 0 h and 24 h, and OPTPro 3000 software was used for photographing and observation, respectively, and was repeated three times. Then the cell migration situation was analyzed by ImageJ-win64 software, and the cell migration speed and mobility were calculated under different conditions. The cell migration speed was equal to the cell migration area divided by the cell migration time, and the cell mobility was equal to the cell migration area divided by the 0 h cell scratch area.

### 2.5. Immunocoprecipitation

After placing different group transfection plasmids into 293T cells, the old medium was discarded and washed with PBS. After that, 200 μL protein lysate was added to each well and the lysate was collected into 1.5 mL centrifuge tubes after full lysis. After 20 min flipping at 4 °C and centrifuging, the supernatant was collected and the protein concentration was determined using the BCA protein concentration assay kit (Beyotime, Beijing, China). We prepared 10 aliquots of 12 μL each sample, quantified into 25 μg, and added them to 5×loading buffer. After 5 min of denaturing at high temperature, the samples were placed at −20 °C and an input control group was created as a reserve. The remaining protein was combined with appropriate number of magnetic beads to prepare the sample group (IP) and negative control group (IgG). The samples were used a magnetic frame to discard the supernatant after low-temperature incubation, and washed with pre-chilled PBS. After that, an appropriate amount of 1× loading buffer was added to the sample group and the negative control group, and they were boiled for 8–10 min to obtain the desired sample for subsequent Western blot experiments.

### 2.6. Western Blot

After placing the SDS-PAGE gel into the swimming tank and adding 1× SDS-PAGE electrophoresis buffer for Tris-Gly, protein samples were added and separated by 80 V electrophoresis for 20 min and 120 V electrophoresis for 65 min. We then cut the gel after electrophoresis according to the position of the marker. After transferring the film, it was closed in a shaker at room temperature for 2 h. The samples were cleaned with 1× TBST 10 min 3 times and incubated at 4 °C for primary antibody. The Flag antibody was obtained from Sigma-Aldrich (Hamburg, Germany) with a catalog number of F1804. The HA antibody was obtained from CellSignaling Technology (Danvers, MA, USA) with a catalog number of 3724T. The dilution ratio of both antibodies was 1:1000. After overnight incubation, we washed them three times for 10 min each time. Then we incubated them with a secondary antibody. The secondary antibody was obtained from Abmart (Shanghai, China) with a dilution ratio of 1:5000. We incubated it on an oscillator for 1 h. Finally, we scanned the target protein bands with a chemiluminescence imager.

### 2.7. The Co-Localization of EDNRB and EDN1 Was Verified by Laser Confocal

The 293T cells were cultured into confocal cell culture dishes, and the plasmids of wild-type EDNRB, mutant-type EDNRB, and EDN1 were co-transfected into 293T cells in pairs. After 24 h of culturing for the co-transfection, the cells were washed with PBS and fixed with 4% paraformaldehyde at room temperature for 15 min. Then, the fixative solution was discarded and washed with PBS. Subsequently, we added PBS containing 1% Triton-X100 for 15 min and blocked with 5% BSA PBS for 1 h. The primary antibody was incubated immediately. The secondary antibody was labeled with AlexaFluor 555 donkey anti-mouse igG (H+L) and AlexaFluor 488 goat anti-rabbit igG indicator proteins, and the nuclei were indicated by DAPI and then observed under confocal microscopy.

## 3. Results

### 3.1. Structural Prediction Showed That There Were Significant Differences in Three-Dimensional Structure Between Wild and Mutant EDNRB Proteins

In order to compare the difference between the three-dimensional structure of wild-type EDNRB and mutant-type EDNRB, the online software SWISS (https://swissmodel.expasy.org/ accessed on 27 August 2023) was used to predict the protein structure. The results showed that the missing segment was 404–443 aa and was significantly different from the wild-type EDNRB protein. In the melanin pathway, EDN1 can activate phospholipase Cγ (*PLCγ*) after binding to its receptor EDNRB. After EDN1 binding to EDNRB, the hydrolysis of phospholipid phosphatidylinositol 4,5-biphosphate (PIP2) to generate inositol triphosphate (IP3) and diacylglycerol (DAG) will be promoted. The IP3 can increase intracellular Ca^+^, while the DAG can induce the activation of protein kinase C (PKC), which activates rapidly accelerated fibrosarcoma (Raf) by phosphorylation. The process leads to the activation of the MAPK cascade, and finally regulates the transcription of MITF to control the synthesis of melanin particles (Figure 1D) [20,21]. It is speculated that the mutant-type EDNRB protein may affect the binding to its ligand, then affect the synthesis of melanin and ultimately lead to the emergence of the TEB phenotype in pigs.

### 3.2. The Mutant EDNRB Significantly Reduced Transcription Levels of Melanin Pathway Genes

Recombinant plasmids of wild-type EDNRB, mutant-type EDNRB, and EDN1 were constructed using p3xFLAG-CMV and pCMV-N-HA vectors, respectively. PCR was performed to verify the target region of the constructed recombinant plasmid. The results showed that the sizes of wild-type EDNRB, mutant-type EDNRB, and EDN1 were 1614 bp, 1497 bp, and 817 bp, respectively. It was suggested that the recombinant plasmids p3xFLAG-CMV-14-EDNRB (Figure 2A), p3xFLAG-CMV-14-EDNRB_M (Figure 2B), and pCMV-N-HA-EDN1 (Figure 2C) were successfully constructed. Then, we divided the wild-type EDNRB, mutant-type EDNRB, and EDN1 plasmids into groups and co-transfected them into B16 cells for 24 h before extracting RNA. We then obtained cDNA through reverse transcription and performed RT-qPCR for the three targeting genes participating in the melanogenesis pathway, including *PLCγ*, *Raf*, and *MITF*. The RT-qPCR results of three repeated experiments showed that after co-transfection of wild-type EDNRB and EDN1, the expression of *PLCγ* showed a significantly higher trend than that in the EDN1 group (Figure 2D). The mean expression of the co-transfected wild-type EDNRB and EDN1 groups was about 8.17-fold that of the EDN1 group. *Raf* and *MITF* expression levels were 1.39-fold and 4.46-fold higher (Figure 2E,F). Compared with the wild-type group, the expressions of *PLCγ*, *Raf*, and *MITF* in the mutant EDNRB group decreased by 2.04-fold, 2.19-fold, and 1.79-fold, respectively (*p*-value < 0.01, *t*-test, Figure 2D–F). The mean expression of *PLCγ* and *MITF* increased by 4-fold and 2.48-fold that of EDN1 group, respectively (Figure 2D,F). Interestingly, the expression of *PLCγ* in the co-transfection mutant EDNRB and EDN1 group was reduced by 1.58-fold that the EDN1 group (Figure 2E). Therefore, it is inferred that mutant-type EDNRB will lead to the decrease in downstream gene expression, so it could affect the *MITF* gene that can regulate melanin expression.

### 3.3. Overexpression Experiment Verified the Mutant EDNRB Decreased the Melanin Content

Compared with the wild-type EDNRB group, the results of three repeated experiments showed that the melanin production in the mutant-type EDNRB and EDN1 co-transfected group was significantly decreased (*p*-value < 0.01, *t*-test) (Figure 3), while the melanin production in the wild-type EDNRB and EDN1 co-transfected group was higher than that of the negative control (NC) group and the mutant-type. The results further indicated that the co-transfection of mutant-type EDNRB and EDN1 not only reduced the expression of *PLCγ*, *Raf*, and *MITF* genes in the melanin pathway, but also affected the production of melanin.

### 3.4. Overexpression Experiments Confirmed That the Mutant EDNRB Affected Melanocyte Migration

Studies have shown that EDNRB affects melanocyte migration and melanin expression during the embryonic period of animals [22]. Considering that the TEB coat color phenotype in pigs is black at both ends and white in the middle (Figure 1A), we speculated that the formation of the TEB coat color in pigs may be related to early melanocyte migration. In order to confirm the hypothesis, three repeated wound-healing experiments were carried out. The results showed that at 0 h, no migration occurred in the control group, the cells carrying wild-type EDNRB, and the mutant-type EDNRB. At the 24th hour, there was a certain degree of migration in the NC control group cells, with an average migration speed of 41,522.1 μm^2^/h. Almost all cells carrying wild-type EDNRB migrated, with an average migration speed of 78,550 μm^2^/h. The cells carrying the mutant-type EDNRB underwent some migration, with an average migration rate of 54,306.3 μm^2^/h (Figure 4A,B). The migration speed of cells carrying mutant-type EDNRB was slightly higher than that of NC control, but significantly lower than that of cells carrying wild-type EDNRB, indicating that mutant-type EDNRB reduced the migration speed of melanocytes (*p*-value < 0.01, *t*-test). To further verify these results, we conducted two replication experiments, which confirmed that cells carrying mutant-type EDNRB reduced melanocyte migration rate (Appendix A). Subsequently, we compared the migration rates of cells carrying wild-type and mutant-type EDNRB. Comparing with the NC control group, cells carrying wild-type EDNRB and mutant-type EDRNB increased the migration rate of melanocytes (Figure 4C). However, the mutant-type EDNRB had a slightly higher cell migration than that of the NC control group, and the wild-type EDNRB had a much higher rate than the NC. In addition, the mutant-type EDNRB was much lower than the wild type. The above results indicate that the mutation of EDNRB may affect the melanocytes’ migration, which implies the prevention of melanocytes migrating to the whole body during the embryonic stage.

### 3.5. Immunocoprecipitation Showed That Mutant EDNRB Could Not Interact with EDN1

We used the 293T cells for cell transfection and conducted co-immunoprecipitation (Co-IP) experiments. After transfecting 293T cells with three recombinant plasmids, including p3xFLAG-CMV-14-EDNRB, p3xFLAG-CMV-14-EDNRB_M, and pCMV-N-HA-EDN1, and extracting protein product, the Western blot assay was performed. The results showed that the three recombinant plasmids successfully expressed proteins of 52 kDa, 48 kDa, and 27 kDa, respectively, indicating that the plasmid proteins constructed by these three genes could normally express (Figure 5A). The results of the immunofluorescence assay (IFA) showed significant red fluorescence in the experimental group transfected with p3xFLAG-CMV-14-EDNRB and p3xFLAG-CMV-14-EDNRB_M (Figure 5B). These results indicate that both wild-type and mutant-type EDNRB could be successfully expressed in 293T cells. The results of Western blot and IFA verified that the recombinant plasmid pCMV-N-HA-EDN1 in 293T cells was successfully expressed as well (Figure 5A,C).

To further confirm whether the protein from mutant-type EDNRB can continue to interact with EDN1, we co-transfected the p3xFLAG-CMV-14-EDNRB, p3xFLAG-CMV-14-EDNRB-M, and pCMV-N-HA-EDN1 eukaryotic expression plasmids into 293T cells in different groups to explore their interaction and co-localization. The results of the immunocoprecipitation experiment showed that the input group in the Co-IP experiment for wild-type EDNRB and EDN1 could express normal bands. The bands in the Flag and HA of the IP group indicated that protein from wild-type EDNRB and EDN1 could normally interact (Figure 5D). However, the FLAG group of mutant-type EDNRB and EDN1 could express successfully, while no bands in HA could, which indicates that the proteins from mutant-type EDNRB and EDN1 cannot interact (Figure 5E). We further transfected p3xFLAG-CMV-14-EDNRB and pCMV-N-HA-EDN1 plasmids into 293T cells. After 24 h of culturing, we performed immunofluorescence experiments on the samples. It was found that proteins of wild-type EDNRB and EDN1 have obvious co-localization phenomena in space under confocal laser microscopy (Figure 5F), which further indicates the interaction between wild-type EDNRB and EDN1.

## 4. Discussion

Phenotypic studies in mice have reported that *EDNRB* plays an important role in the development of myoenteric ganglion neurons and epidermal melanocytes [23]. The EDNRB-mediated signaling pathway is necessary for the migration phase of melanoblast and enteric neuroblast development [24]. The pigment deficiency is localized to skin melanocytes, suggesting that the coat color spots are due to disruption in the development of neural crest-derived melanocyte precursors, rather than due to a defect in the ability to produce melanin [25]. Specific ligand or receptor mutations cause dysfunction, which will affect skin pigmentation [26]. Further research has shown that mutations in the *EDNRB* gene can result in almost completely white skin and fur in mice, typically only with small, clearly defined black areas on the head and tail [27,28].

EDNRB, as a G protein coupled receptor with seven transmembrane regions, has high affinity for EDN1, EDN2, and EDN3 ligands. Studies have shown that ultraviolet light can cause keratinous cells to continuously secrete EDN1 and EDN3, and their binding to EDNRB will accelerate pigment synthesis, indicating that both ligands have the ability to increase melanin expression [28]. Proteins of EDNRB interacting with EDN1 will promote epidermal melanocyte production, pigmentation in normal skin, and MC1R-deficient skin in melanocytes (McSC) [29]. Therefore, we selected the melanin pathway including EDN1 on the basis of this study to verify the effect of EDNRB mutation on melanin production.

The results showed that the TEB coat color in Chinese pigs was affected by a major gene *EDNRB* and multiple modified gene loci (*KIT*, *KITLG*, etc.) [30,31,32]. *MITF* and *EDNRB* are subject to strong artificial selection in Chinese white spotted pigs, different white spotted breeds may have different *MITF* and *EDNRB* mutant alleles, and the interaction between them leads to different regulatory effects and forms different color patterns [33]. Further analysis showed that one 11 bp deletion in *EDNRB* gene (chromosome 11: 50,076,945–50,076,960 bp) was a candidate causal mutation for the Chinese pigs with TEB coat color, except for Jinhua pigs [19]. This mutation introduces a premature stop codon, truncating the EDNRB protein and impairing its function and resulting in the formation of TEB pigs. In addition, researchers found that EDN1 binding to its receptor EDNRB can activate a series of signaling pathways [34]. In order to confirm how mutant EDNRB affects the expression of melanin in cells, we selected three key genes participating in the melanin pathway and detected their expression, including *PLCγ*, *Raf*, and *MITF*. We explored the effect of mutant-type *EDNRB* on melanocytes through RT-PCR, wound-healing, and other experiments. Based on the results of previous studies and speculation, our study demonstrated the difference between mutant-type *EDNRB* and wild-type *EDRNB* by predicting the proteins of the three-dimensional structure. Promoting the up-regulation of *EDNRB* gene will increase the expression of *MITF* [35]. In this study, the expression levels of *MITF* and two key genes downstream of *EDNRB*, namely, *PLCγ* and *Raf*, were detected by RT-qPCR. We constructed wild-type and mutant recombinant plasmids and transfected them into mouse B16 melanoma cells. We then performed an RT-qPCR experiment on the genes participating in the melanin pathway and detected the melanin expression level. The results showed that the expression levels of *PLCγ*, *Raf*, and *MITF* in the mutant-type *EDNRB* cells were significantly reduced, and the melanin expression level was also significantly decreased compared with the wild-type *EDNRB* cells. It is speculated that the early termination of protein translation caused by the 11 bp mutation of *EDNRB* will lead to the loss of the function.

The formation of the TEB phenotype is closely related not only to melanin and related genes expression but also to melanocyte migration, which may be a potential contributing factor. Therefore, in this study, we wanted to explore the effect of mutant-type *EDNRB* on melanocyte migration by wound-healing experiment [36,37]. The *EDNRB* can promote the migration ability of melanocytes, and its mutations can directly affect the phenotype in animals, which is consistent with the discovery that early migrating neural crest cells can widely express *EDNRB* [28]. In this study, the effect of mutant-type *EDNRB* on melanocyte migration was verified by comparing the migration capacity of cells carrying mutant-type and wild-type *EDNRB*. Our results showed that cells with mutant-type *EDNRB* significantly reduced the migration ability compared with that of the wild type, and were close to the NC control group. The mutant *EDNRB* is inferred to affect melanocyte migration during the embryonic period, preventing melanocytes from migrating to the body, which ultimately results in the TEB phenotype in pigs. Thus, the abnormal melanocyte migration may cause the TEB coat color phenotype in Chinese pigs. However, further research is needed to determine which stage of the embryo influences cell migration. In addition, the fact that the two ends (head and tail) of the melanocytes can product normal melanin but the body cannot still needs to be deeply explored. The coat color of animals plays an important role in livestock breeding. For example, an indel of *MITF* causes white feathers in Beijing ducks [38]. In addition, mutant *EDNRB* not only affects animal pigmentation, but also affect human health. For example, *EDNRB* and *SOX10* mutations together cause human Waardenburg syndrome [3].

Previous studies pointed out that abnormal melanin deposition occurs after *EDNRB* mutation in mice. The researchers speculated that the failure of normal melanocyte and intestinal neuron development was caused by the inability of EDN3 to interact with the mutant EDNRB [39]. In order to further explore the protein interaction of the melanin pathway, we conducted the interaction between proteins of wild-type EDNRB and mutant-type EDNRB and EDN1 proteins, respectively. The results showed that proteins of mutant-type *EDNRB* could not interact with EDN1, which further indicated that mutant-type *EDNRB* had an impact on melanin pathway. Moreover, the difference between wild-type and mutant-type EDNRB was predicted according to the three-dimensional structure in the experiment, and it was found that the mutant EDNRB protein lacks the 404–443 amino acid segment, which likely disrupts ligand binding and downstream signaling. Combined with the results of the Co-IP experiment, we inferred that the binding region of EDNRB and EDN1 was 404–443 aa, and the mutant EDNRB was precisely due to the absence of this sequence. It failed to induce the expression of *PLCγ*, *Raf*, *MITF*, and other genes in the melanin pathway, and ultimately reduced the production of melanin in cells. However, this is only a speculation at present, and the mechanism needs further study.

## 5. Conclusions

Our study found that mutant-type *EDNRB* significantly reduced the expression of genes participating in the melanin pathway and it ultimately affected the production of melanin. At the same time, we also confirmed that mutant-type *EDNRB* reduced the migration rate and cell mobility of melanocytes, suggesting that mutant-type *EDNRB* may have begun to affect the migration of melanophore cells during the embryonic period. In addition, the Co-IP assay further confirmed that the mutant-type EDNRB protein could not bind to the EDN1 protein normally. In summary, our study revealed the possible molecular mechanism of the 11 bp deletion in *EDNRB* affecting the formation of TEB coat color phenotype in Chinese local pigs.

## Figures and Tables

**Figure 1 animals-15-00478-f001:**
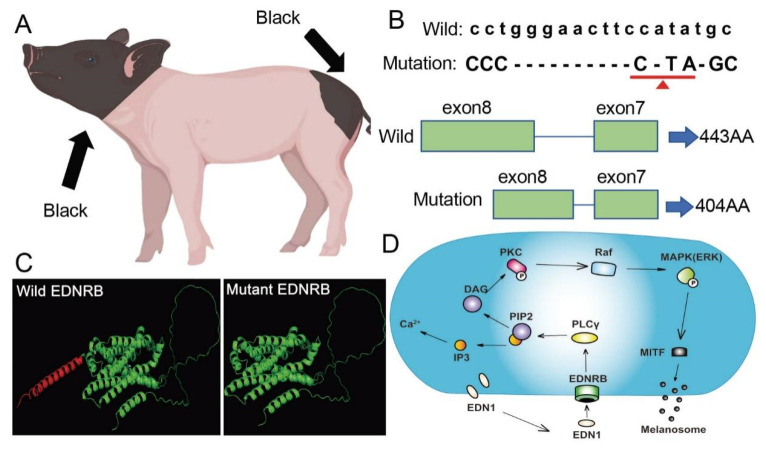
Two-end black (TEB) coat color phenotype, schematic diagram of EDNRB transcripts and their encoded protein, three-dimensional structure prediction of the EDNRB protein, and an EDNRB melanin pathway diagram. (**A**) A diagram of the TEB pig phenotype. (**B**) The red inverted triangle indicates that the 11 bp indel caused premature stop codon of the EDNRB in TEB pigs. (**C**) A three-dimensional structural prediction diagram of the wild (**left**) and mutant (**right**) EDNRB proteins. (**D**) The *EDNRB*-associated melanin signaling pathway. PLCγ: phospholipase Cγ; PIP2: phosphatidylinositol 4,5-biphosphate; IP3: inositol triphosphate; DAG: diacylglycerol; PKC: protein kinase C; P: phosphorylation; Raf: rapidly accelerated fibrosarcoma; ERK: extracellular regulated protein kinases; MAPK: mitogen-activated protein kinase; MITF: microphthalmia-associated transcription factor.

**Figure 2 animals-15-00478-f002:**
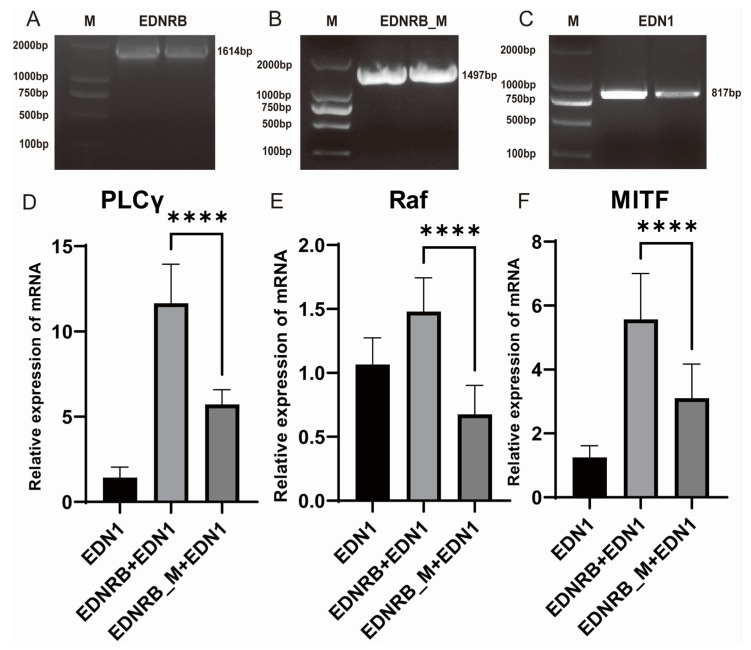
Overexpression of recombinant plasmids, expression of *PLCγ*, *Raf*, and *MITF* genes in B16 cells transfected with plasmids of different groups, and melanin expression detection. (**A**,**B**) Overexpression of wild and mutant *EDNRB*. (**C**) Overexpression of *EDN1*. (**D**–**F**) The expression levels of *PLCγ*, *Raf*, and *MITF* in B16 cells, respectively. The **** above the bars indicate significant differences at *p*-value < 0.01.

**Figure 3 animals-15-00478-f003:**
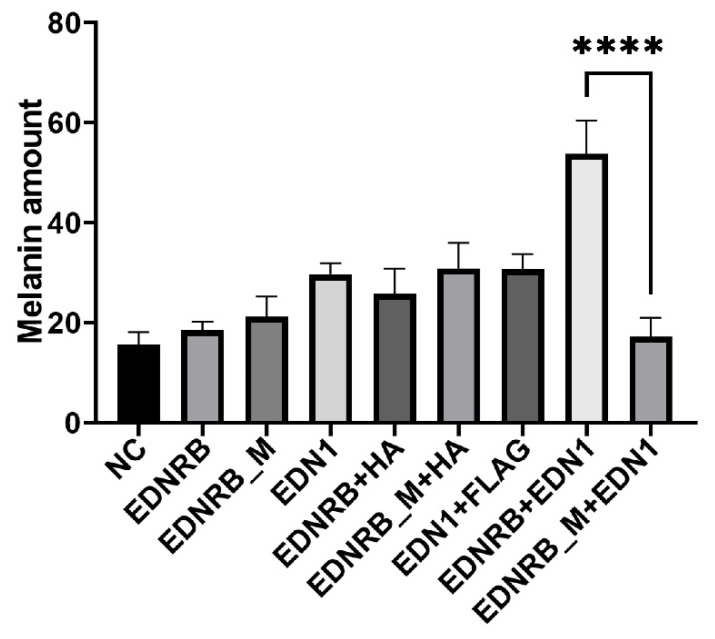
Melanin expression detection. Changes in melanin content in B16 cells after transfection with different grouping plasmids. NC: negative control; EDNRB_M: mutant EDNRB; HA: pCMV-N-HA; Flag: p3xFLAG-CMV. The EDNRB, EDNRB_M, and EDN1 groups were used to analyze the expression of melanin when transfected with a single plasmid; the addition of Flag and HA plasmid groups was used to exclude the effect of plasmid vector; the EDNRB+EDN1 and EDNRB_M + EDN1 groups were used to verify the effect of mutant EDNRB on the expression of melanin in the case of overexpression. The **** above the bars indicate significant differences at *p*-value < 0.01.

**Figure 4 animals-15-00478-f004:**
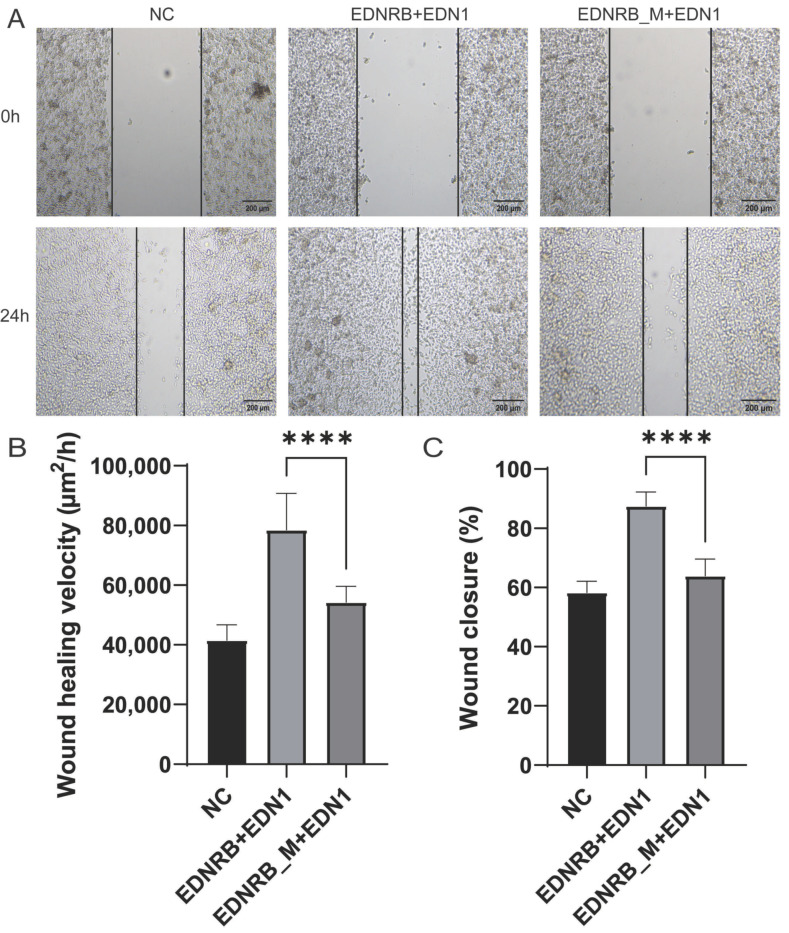
Cell migration rate and cell mobility detection. (**A**) Cell migration maps, including control, wild-type *EDNRB* and *EDN1* groups, and *EDNRB_M* and *EDN1* groups. M: 2000 bp DNA ladder marker. (**B**) Cell migration velocity. (**C**) Cell migration. The **** above the bars indicate significant differences at *p*-value < 0.01.

**Figure 5 animals-15-00478-f005:**
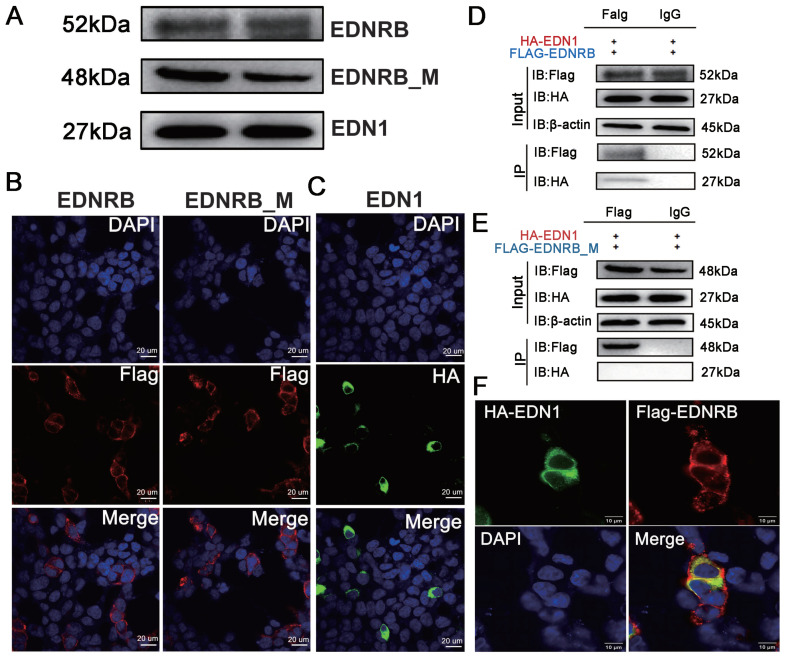
Recombinant plasmids were overexpressed; co-immunoprecipitation of wild and mutant EDNRB protein, and immunofluorescence of wild-EDNRB. (**A**) Protein expression after transfection of wild-*EDNRB*, mutant *EDNRB*, and *EDN1* into 293T cells. EDNRB: wild-type *EDNRB*; EDNRB_M: mutant *EDNRB*. (**B**) The immunofluorescence map of wild and mutant EDNRB. (**C**) The immunofluorescence map of *EDN1*. (**D**) The co-immunoprecipitation experiment between wild-EDNRB and EDN1 protein. (**E**) The co-immunoprecipitation between mutant EDNRB and EDN1 protein. (**F**) The immunofluorescence co-localization between wild-EDNRB and EDN1 protein.

## Data Availability

None of the data were deposited in an official repository. Data are available upon request. Data may be made available to the corresponding author upon reasonable request during the study period.

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
