# Peer review of "The Impact of Mutant *EDNRB* on the Two-End Black Coat Color Phenotype in Chinese Local Pigs"

_animals, 2025, doi:10.3390/ani15040478_

Round 1

Reviewer 1 Report

Comments and Suggestions for Authors

Dear Author,

I commend your efforts on this study, which presents significant insights into the genetic basis of coat color phenotypes in Chinese pigs. The research is scientifically sound, and the experiments are well-designed and executed. However, to improve the clarity, reproducibility, and impact of your manuscript, I suggest addressing the following minor revisions:

Materials and Methods Section

Some procedural details, such as specific concentrations, incubation times, or reaction conditions, are either missing or vaguely described.

Example: "The cells were then cultured in an incubator with 37°C and 5% CO2 for 24 h." This does not specify whether media changes or other conditions were monitored during this period.

Recommendation: Provide exact reagent concentrations, incubation times, and conditions for each step to ensure reproducibility.

Some phrases are inconsistently or vaguely worded, making it difficult to replicate experiments.

Example Correction: Replace “co-transfected into B16 cells” with “Plasmids p3xFLAG-CMV-14-EDNRB and pCMV-N-HA-EDN1 were co-transfected into B16 cells using Lipofectamine 3000, following the manufacturer’s protocol.”

Original: “The construction of these plasmids was completed by a biological company (Tsingke Biological Company, China).”

Corrected: “Plasmid construction was performed by Tsingke Biological Company (China).”

Missing information about key reagents, such as the composition of lysis buffer or PCR cycling conditions.

Example Correction: “200 μL of the prepared lysis buffer” should specify the buffer composition, e.g., “200 μL lysis buffer (50 mM Tris-HCl, 150 mM NaCl, 1% Triton X-100, pH 7.4).”

Comments:

Plasmid Construction: This section is generally clear but lacks details about the design process, including how sequences were selected or validated.

Gene Expression Analysis: Provides a reasonable overview but omits RT-qPCR cycling conditions and primer efficiencies, which are critical for reproducibility.

Melanin Quantification: The description of ELISA lacks specific details on controls and standards used for calibration.

Cell Migration Assay: While the scratch assay method is standard, details like the imaging system used or criteria for defining “migration area” are not provided.

Immunoprecipitation and Western Blot: These sections need more specifics, such as the antibodies used, their dilutions, and blocking conditions.

Example Improvements

Original:

“After cell culture for 24 h, cells were washed with PBS and marked in the petri dish by a 10 μL pipette.”

Revised:

“After culturing cells for 24 hours in a 6-well plate, a scratch was created using a 10-μL pipette tip. Cells were washed twice with PBS to remove debris, and fresh DMEM medium without FBS was added.”

The Materials and Methods section is thorough but requires more precise and detailed descriptions to ensure reproducibility. With the recommended changes, this section will better serve as a resource for other researchers seeking to replicate the study.

Results Section

Improvements:

Streamlining Redundant Text:

Many sentences in the results section are repetitive. For example, the expression results of PLCγ, Raf, and MITF are repeated in different forms within the same paragraph.

Example:

Original: "The mean expression of PLCγ showed a significantly higher trend than that in the negative control group (Negative Control, NC). The mean expression of the co-transfected wild-type EDNRB and EDN1 groups was about 4.51 times than that of the NC group."

Streamlined: "The expression of PLCγ in the wild-type EDNRB and EDN1 co-transfected group was 4.51-fold higher than the negative control (NC)."

Detailed Quantitative Reporting:

While fold changes are reported, exact P-values and statistical tests are often omitted or only vaguely mentioned (e.g., “reached an extremely significant level”).

Recommendation: Always provide exact statistical values (e.g., P = 0.001) and name the statistical tests used (e.g., t-test, ANOVA).

Recommendations:

Simplify and Consolidate Results:

Group similar results (e.g., gene expression findings) into a single cohesive statement instead of describing each gene separately in detail.

Example:

Original: "The mean expression of Raf was about 1.79 times higher than that of the NC group, and the mean expression of MITF was 4.02 times higher than that of the NC group."

Revised: "Raf and MITF expression levels were 1.79-fold and 4.02-fold higher, respectively, in the wild-type EDNRB group compared to NC."

Report Statistical Tests and Values:

Specify the statistical tests used and include exact P-values.

Example Correction: Instead of saying “extremely significant level,” state: “The increase in PLCγ expression in the wild-type EDNRB group was statistically significant (P < 0.001, t-test).”

Corrections:

Grammatical and Syntax Issues:

Original: "We then obtain cDNA by reversing transcription."

Corrected: "We then obtained cDNA through reverse transcription."

Terminology Consistency:

Ensure consistency in referring to genes and proteins (e.g., PLCγ vs. Plcγ).

Redundant Sentences:

Original: "The expression of PLCγ, Raf, and MITF was much lower than that of the wild type, which reached an extremely significant level."

Corrected: "The expression of PLCγ, Raf, and MITF was significantly lower in the mutant EDNRB group compared to the wild-type group (P < 0.01)."

Comments:

The Results section effectively covers all key experiments and findings, but redundancy and a lack of statistical details reduce clarity.

Discussion Section

Improvements:

Reduce Repetition:

Several points, such as the effect of the 11-bp deletion on protein truncation and melanocyte migration, are repeated multiple times.

Example:

Original: "The 11-bp mutation causes the EDNRB to produce a stop codon in advance, which truncates the EDNRB protein, thus affecting the function of EDNRB."

Revised: "The 11-bp mutation introduces a premature stop codon, truncating the EDNRB protein and impairing its function."

Expand on Broader Implications:

The discussion briefly mentions the relevance of the findings to pigmentation genetics but does not delve into potential applications for breeding or insights into other species.

Recommendation: Discuss how these results can inform genetic selection for coat color in livestock or serve as a model for studying human pigmentation disorders.

Propose Future Research:

The discussion mentions the need for further studies but does not suggest specific experimental approaches.

Recommendations:

Organize the Discussion into Subsections:

Use headings like "Mechanistic Insights," "Comparison with Previous Studies," "Broader Implications," and "Future Directions" to improve readability and structure.

Address Unresolved Questions:

Propose specific hypotheses for why melanin is produced only in the head and tail regions but not the body.

Example Suggestion: "Future studies could explore whether region-specific expression of other pigmentation  genes (e.g., KIT or KITLG) contributes to the observed pattern."

Add Supporting References:

Some speculative claims, such as the exact role of the 404-443 region in EDNRB-EDN1 binding, lack supporting references. Include citations to relevant structural biology or pigmentation studies.

Corrections:

Grammar and Syntax Issues:

Original: "Further researches has shown that defects in the EDNRB gene..."

Corrected: "Further research has shown that defects in the EDNRB gene..."

Clarify Speculative Statements:

Original: "The end of mutant EDNRB was truncated amino acid polypeptide compared with that of wild type."

Corrected: "The mutant EDNRB protein lacks the 404-443 amino acid segment, which likely disrupts ligand binding and downstream signaling."

Avoid Repetition:

Original: "The above results confirmed that mutant-type EDNRB could indeed reduce the migration rate of melanocytes."

Corrected: "These results confirm that mutant-type EDNRB reduces melanocyte migration."

Comments:

The Discussion section provides a strong link between the study findings and prior research but would benefit from a deeper exploration of the broader implications.

Repetition detracts from the readability and impact of the discussion.

Future directions need to be more specific to guide subsequent research.

Author Response

Reviewer #1: I commend your efforts on this study, which presents significant insights into the genetic basis of coat color phenotypes in Chinese pigs. The research is scientifically sound, and the experiments are well-designed and executed. However, to improve the clarity, reproducibility, and impact of your manuscript, I suggest addressing the following minor revisions:
Response: We are very grateful for your professional comments on our article. As you are concerned, there are several issues that need to be addressed. According to your constructive comments, we have made extensive corrections in our revised manuscript. The changes we made to the revised manuscript were marked in red font. We hope that the revised manuscript will meet your requirements. Our responses to each question as follow:

Materials and Methods Section
1. Some procedural details, such as specific concentrations, incubation times, or reaction conditions, are either missing or vaguely described.
Example: "The cells were then cultured in an incubator with 37°C and 5% CO2 for 24 h." This does not specify whether media changes or other conditions were monitored during this period.
Recommendation: Provide exact reagent concentrations, incubation times, and conditions for each step to ensure reproducibility.
Response: Thank you for your valuable comments. We have modified the procedural details according to your suggestion and also added a detailed description in the revised manuscript by sentences of “The cells were then cultured in incubator with 37 °C and 5% CO2 incubator and cultured for 24 hours using the prepared 1640 complete medium. During this period, the color change of the culture medium was observed. If the color turned yellow, new culture medium could be considered” (lines 120-123).

2.Some phrases are inconsistently or vaguely worded, making it difficult to replicate experiments.
Example Correction: Replace “co-transfected into B16 cells” with “Plasmids p3xFLAG-CMV-14-EDNRB and pCMV-N-HA-EDN1 were co-transfected into B16 cells using Lipofectamine 3000, following the manufacturer’s protocol.”
Original: “The construction of these plasmids was completed by a biological company (Tsingke Biological Company, China).”
Corrected: “Plasmid construction was performed by Tsingke Biological Company (China).”
Missing information about key reagents, such as the composition of lysis buffer or PCR cycling conditions.
Example Correction: “200 μL of the prepared lysis buffer” should specify the buffer composition, e.g., “200 μL lysis buffer (50 mM Tris-HCl, 150 mM NaCl, 1% Triton X-100, pH 7.4).”
Response: Thank you for your carefully reviewing. We have corrected the manuscript according to your suggestions. We have described the reagent information in detail, including the brand of lysis buffer and the ratio of protease inhibitor mixture added to the lysis buffer. According to your suggestions, we have provided the brand and product number of the enzyme used in the PCR in the revised manuscript. The relevant sentences are “Plasmid construction was performed by Tsingke Biological Company (China)” (lines 105-106) and “Plasmids p3xFLAG-CMV-14-EDNRB and pCMV-N-HA-EDN1 were co-transfected into B16 cells using Lipofectamine 3000, following the manufacturer’s protocol. The cells were washed with PBS to remove the residual medium, and add 200 μL of Beyotime RIPA Lysis Buffer (Strong) (China) to each well. Among them, RIPA Lysis Buffer needs to add with Beyotime's Protease inhibitor cocktail for general use at a ratio of 1:50” (lines 136-141); “PCR amplification was carried out as follows: denaturation of 98 °C for 10s, followed by 35 cycles of 60 °C for 40s, and a specific Extension temperature of 60 °C for 10s” (lines 113-115).

3.Comments: Plasmid Construction: This section is generally clear but lacks details about the design process, including how sequences were selected or validated.
Gene Expression Analysis: Provides a reasonable overview but omits RT-qPCR cycling conditions and primer efficiencies, which are critical for reproducibility.
Melanin Quantification: The description of ELISA lacks specific details on controls and standards used for calibration.
Cell Migration Assay: While the scratch assay method is standard, details like the imaging system used or criteria for defining “migration area” are not provided.
Immunoprecipitation and Western Blot: These sections need more specifics, such as the antibodies used, their dilutions, and blocking conditions.
Example Improvements
Original:
“After cell culture for 24 h, cells were washed with PBS and marked in the petri dish by a 10 μL pipette.”
Revised:
“After culturing cells for 24 hours in a 6-well plate, a scratch was created using a 10-μL pipette tip. Cells were washed twice with PBS to remove debris, and fresh DMEM medium without FBS was added.”
The Materials and Methods section is thorough but requires more precise and detailed descriptions to ensure reproducibility. With the recommended changes, this section will better serve as a resource for other researchers seeking to replicate the study.
Response: Thank you for your valuable comments. However, since we directly synthesize genes through biological companies for plasmid construction, the design process may be slightly simpler. According to your suggestions, we have added the RT-qPCR cycling conditions by sentences of “Initial denaturation of 95 °C for 20s, followed by 40 cycles of 95 °C for 3s, and a specific annealing temperature of 60 °C for 30s” (lines 131-133). We strictly follow the procedures in the instructions of this brand for RT-qPCR. We also added descriptions of the ELISA and cell migration assay issues you raised in the article, and expanded the details of the Western Blot. The relevant sentences are: “Subsequently, TOYOBO's kod OneTM PCR Master Mix-Blue- item number KMM-201 was used. According to the reference system provided in the instruction manual, the bacterial solution, the universal primers provided by Tsingke, deionized water and PCR enzyme were mixed, and after preparing the mixed solution, PCR was performed on the bacterial solution to verify whether the sequence was successfully constructed. At the same time, the bacterial solution was shaken and expanded, and the cultured bacterial solution was sent to Tsingke Biological Company (China) for sequencing” (lines 105-113); “The mouse melanin ELISA kit provided by Basediao Biotechnology Company Limited (China) was used to detect the cellular melanin content. According to the instructions, first dilute the standard provided in the kit as required, then set up blank wells (no sample and enzyme-labeled reagent, the rest of the steps are the same), standard wells and sample wells to be tested, add 50 μL of sample to be tested to the standard wells, add 40 μL of sample diluent to the sample wells to be tested, and then add 10 μL of sample to be tested (the final dilution of the sample is 5 times), and then place it in a 37 °C incubator for incubation, followed by 5 washes with the washing solution in the kit, and then add 50 μL of enzyme-labeled reagent to each well (except the blank well), repeat the incubation and washing again, and then use the color developer to develop at 37 °C in the dark for 10 minutes, add 50 μL of stop solution to terminate the reaction (the color turns from blue to yellow at this time), and after obtaining the final product, adjust the blank well to zero” (lines 144-156); “After culturing cells for 24 hours in a 6-well plate, a scratch was created using a 10 μL pipette tip. Cells were washed twice with PBS to remove debris, and fresh Dulbecco's Modified Eagle Medium (DMEM) without FBS was added. An inverted biological microscope was used for observation at 0 h and 24 h, and OPTPro software was used for photographing and observation” (lines 163-167); and “The Flag antibody is from Sigma-Aldrich (Germany) with a catalog number of F1804. The HA antibody is from CellSignaling Technology (America) with a catalog number of 3724T. The dilution ratio of both antibodies is 1:1000. After overnight incubation, wash three times for 10 minutes each time. Then incubate with secondary antibody. The secondary antibody is from Abmart (China) with a dilution ratio of 1:5000. Incubate on an oscillator for 1h. Finally, scan the target protein bands with a chemiluminescence imager” (lines 193-199).

Results Section
4.Improvements:
Streamlining Redundant Text:
Many sentences in the results section are repetitive. For example, the expression results of PLCγ, Raf, and MITF are repeated in different forms within the same paragraph.
Example:
Original: "The mean expression of PLCγ showed a significantly higher trend than that in the negative control group (Negative Control, NC). The mean expression of the co-transfected wild-type EDNRB and EDN1 groups was about 4.51 times than that of the NC group."
Streamlined: "The expression of PLCγ in the wild-type EDNRB and EDN1 co-transfected group was 4.51-fold higher than the negative control (NC)."
Detailed Quantitative Reporting:
While fold changes are reported, exact P-values and statistical tests are often omitted or only vaguely mentioned (e.g., “reached an extremely significant level”).
Recommendation: Always provide exact statistical values (e.g., P = 0.001) and name the statistical tests used (e.g., t-test, ANOVA).
Response: Thank you very much for your careful reading of our manuscript. According to your request, we have changed it to (P-value < 0.001, t-test). However, Graphpad software showed P-value < 0.001 and cannot display the exact value. We have changed the NC control group in the figure to EDN1 as the control group, and modified the content of in the revised manuscript as follows: “The mean expression of the co-transfected wild-type EDNRB and EDN1 groups was about 8.17-fold than that of the EDN1 group. Raf and MITF expression levels were 1.39-fold and 4.46-fold higher (Figure 2E, 2F). Compared with the wild-type group, the expressions of PLCγ, Raf, and MITF in the mutant EDNRB group decreased by 2.04-fold, 2.19-fold and 1.79-fold, respectively (P-value < 0.01, t-test, Figure 2D, 2E, and 2F). The mean expression of PLCγ and MITF increased 4-fold and 2.48-fold than that of EDN1 group, respectively (Figure 2D, 2F). Interestingly, the expression of PLCγ in the co-transfection mutant EDNRB and EDN1 group was reduced by 1.58-fold compared with the EDN1 group. (Figure 2E). Therefore, it is inferred that mutant-type EDNRB will lead to the decrease of downstream gene expression, so it could affect the MITF gene that can regulate melanin expression” (lines 241-251).

5.Recommendations: 
Simplify and Consolidate Results:
Group similar results (e.g., gene expression findings) into a single cohesive statement instead of describing each gene separately in detail.
Example:
Original: "The mean expression of Raf was about 1.79 times higher than that of the NC group, and the mean expression of MITF was 4.02 times higher than that of the NC group."
Revised: "Raf and MITF expression levels were 1.79-fold and 4.02-fold higher, respectively, in the wild-type EDNRB group compared to NC."
Report Statistical Tests and Values:
Specify the statistical tests used and include exact P-values.
Example Correction: Instead of saying “extremely significant level,” state: “The increase in PLCγ expression in the wild-type EDNRB group was statistically significant (P < 0.001, t-test).”
Response: Thank you for your valuable suggestions. We have simplified and integrated the manuscript content as you requested. However, the Graphpad can only show the remarkable P-value < 0.001, which means we really difficult to provide the exact P-value.

6.Corrections:
Grammatical and Syntax Issues:
Original: "We then obtain cDNA by reversing transcription."
Corrected: "We then obtained cDNA through reverse transcription."
Terminology Consistency:
Ensure consistency in referring to genes and proteins (e.g., PLCγ vs. Plcγ).
Redundant Sentences:
Original: "The expression of PLCγ, Raf, and MITF was much lower than that of the wild type, which reached an extremely significant level."
Corrected: "The expression of PLCγ, Raf, and MITF was significantly lower in the mutant EDNRB group compared to the wild-type group (P < 0.01)."
Comments:
The Results section effectively covers all key experiments and findings, but redundancy and a lack of statistical details reduce clarity.
Response: Thank you for your correction. We have corrected the grammar and the terminology has been uniformly changed to PLCγ in the revised manuscript. The redundant sentences have also been modified as you requested.

Discussion Section
7.Improvements:
Reduce Repetition:
Several points, such as the effect of the 11-bp deletion on protein truncation and melanocyte migration, are repeated multiple times.
Example:
Original: "The 11-bp mutation causes the EDNRB to produce a stop codon in advance, which truncates the EDNRB protein, thus affecting the function of EDNRB."
Revised: "The 11-bp mutation introduces a premature stop codon, truncating the EDNRB protein and impairing its function."
Response: Thank you for your corrections. We have reduced the redundancy sentences in the revised manuscript.

8.Expand on Broader Implications:
The discussion briefly mentions the relevance of the findings to pigmentation genetics but does not delve into potential applications for breeding or insights into other species.
Recommendation: Discuss how these results can inform genetic selection for coat color in livestock or serve as a model for studying human pigmentation disorders.
Propose Future Research:
The discussion mentions the need for further studies but does not suggest specific experimental approaches.
Response: Thank you for your valuable comments. We have added content in the discussion section based on your advices. The specific additions are as follows: “Coat color of animals plays an important role in livestock breeding. For example, an InDel of MITF causes white feathers in Beijing ducks. In addition, mutant EDNRB not only affects animal pigmentation, but also affect human health. For example, EDNRB and SOX10 mutations together cause human Waardenburg syndrome” (lines 422-426).

9.Recommendations:
Organize the Discussion into Subsections:
Use headings like "Mechanistic Insights," "Comparison with Previous Studies," "Broader Implications," and "Future Directions" to improve readability and structure.
Address Unresolved Questions:
Propose specific hypotheses for why melanin is produced only in the head and tail regions but not the body.
Example Suggestion: "Future studies could explore whether region-specific expression of other pigmentation genes (e.g., KIT or KITLG) contributes to the observed pattern."
Response: Thank you for your valuable suggestions. We have added our hypotheses about the TEB phenotype in pig and described by “The mutant EDNRB is inferred to affect melanocyte migration during the embryonic period, preventing melanocytes from migrating to the body, which ultimately results in the TEB phenotype in pigs” (lines 416-418).

10.Add Supporting References:
Some speculative claims, such as the exact role of the 404-443 region in EDNRB-EDN1 binding, lack supporting references. Include citations to relevant structural biology or pigmentation studies.
Response: Thank you for your valuable comments. We have not found any literature on the exact role of EDNRB-EDN1 binding in the 404-443 region, so the role of this region remains to be studied. However, we cited in the 39th reference that EDNRB lost its amino acids due to mutation and could not bind to EDN3, which eventually made it lose its function. The 29th reference explained that EDNRB has affinity for EDN1, EDN2, and EDN3, and elaborated on its related structural biology. The 30th reference revealed that EDNRB binding to EDN1 promotes melanin deposition. I hope the above explanations will satisfy you.

11. Corrections:
Grammar and Syntax Issues:
Original: "Further researches has shown that defects in the EDNRB gene..."
Corrected: "Further research has shown that defects in the EDNRB gene..."
Clarify Speculative Statements:
Original: "The end of mutant EDNRB was truncated amino acid polypeptide compared with that of wild type."
Corrected: "The mutant EDNRB protein lacks the 404-443 amino acid segment, which likely disrupts ligand binding and downstream signaling."
Avoid Repetition:
Original: "The above results confirmed that mutant-type EDNRB could indeed reduce the migration rate of melanocytes."
Corrected: "These results confirm that mutant-type EDNRB reduces melanocyte migration."
Response: Thank you for your careful review. We have made corrections according to your comments and hope you will be satisfied.

12. Comments: The Discussion section provides a strong link between the study findings and prior research but would benefit from a deeper exploration of the broader implications. Repetition detracts from the readability and impact of the discussion. Future directions need to be more specific to guide subsequent research.
Response: Thank you for your review and valuable feedback on our paper. After careful consideration, we have added a small section to answer the questions, and we hope that these improvements will further improve the paper. The specific additions are as follows: “Coat color of animals plays an important role in livestock breeding. For example, an InDel of MITF causes white feathers in Beijing ducks. In addition, mutant EDNRB not only affects animal pigmentation, but also affect human health. For example, EDNRB and SOX10 mutations together cause human Waardenburg syndrome” (lines 422-426).

Reviewer 2 Report

Comments and Suggestions for Authors

The authors constructed mutant and wild type EDNRB and demonstrated mutant EDNRB 1) cannot induce melanin synthesis pathway; 2) inhibit melanin production; 3) reduces melanocyte migration; 4) cannot bind to its ligand. The results can be an explanation to the phenotype of TEB pig. I think the experiment results are solid and believable and the conclusion provides insight to the understanding the pathway. There are some issues I think need to be clarified or revised.

If I understand it correctly, the mutant EDNRB protein lacks 40aa, which contains “two helical transmembrane regions and four protein-binding regions”. It may be too many regions in just 40aa. Please label these regions (or domains) on the missing peptide.   

“the expression of Raf in the mutant-type EDNRB was even lower than that of the NC group (Figure 2E).” Please provide statistical significance analysis between NC and mutant ednrb. And the conclusion “mutant-type EDNRB will lead to the decrease of downstream gene expression” is lacking support.

In figure2, I think there should be a control EDN1 only, for comparison to the two experimental groups.

Please add statistical analysis to the method section, or at least indicate the meaning of asterisks in the figure legends.

There are many tiny issues, for example, “wild EDNRB” should be “wild-type EDNRB”.

Line 29-30: “expression of the three genes in mutant EDNRB cells” should be “expression of the three genes in cell line expressing mutant EDNRB

In the discussion, mutant EDNRB “can’t bind to 395 EDN1, and then inhibit the expression of PLCγ, Raf, MITF, and other genes in the melanin 396 pathway”. In my understanding, loss of binding doesn’t necessarily lead to inhibit downstream gene expression. “failed to induce” would be more appropriate.

Figure2 y axis “Elevate” should be “Relative”

Format issues, for example in 3.2, gene names, construct names and mRNA should be Italic.

Figure3 and the according text, it’s the melanin getting measure rather than “melanin gene”, so it should be melanin production instead of melanin expression.

Line 212, tense “obtain”

Line 238, lowercase “while”

Figure5A, lane miss labeling

Comments on the Quality of English Language

I don't have problem understanding the context and the manuscript is very readable. There are some small problems like mis-used tense and I think authors need to be more careful on using terminology. 

Author Response

Reviewer #2: The authors constructed mutant and wild type EDNRB and demonstrated mutant EDNRB 1) cannot induce melanin synthesis pathway; 2) inhibit melanin production; 3) reduces melanocyte migration; 4) cannot bind to its ligand. The results can be an explanation to the phenotype of TEB pig. I think the experiment results are solid and believable and the conclusion provides insight to the understanding the pathway. There are some issues I think need to be clarified or revised.
Response: Thank you very much for your valuable comments. Your comments are very valuable and helpful to us. We have carefully read the opinions and made modifications. The specific corrections are as follows.

1.    If I understand it correctly, the mutant EDNRB protein lacks 40aa, which contains “two helical transmembrane regions and four protein-binding regions”. It may be too many regions in just 40aa. Please label these regions (or domains) on the missing peptide.
Response: Thank you for your careful reading and giving us professional advice. Since we did not do a corresponding study of helical transmembrane regions and four protein-binding regions here, we omitted a description of this problem from the revised manuscript.

2.    “the expression of Raf in the mutant-type EDNRB was even lower than that of the NC group (Figure 2E).” Please provide statistical significance analysis between NC and mutant EDNRB. And the conclusion “mutant-type EDNRB will lead to the decrease of downstream gene expression” is lacking support.
Response: Thank you very much for pointing out the problem. We have made changes to it. We have used EDN1 as the control group and have obtained the reduced value through statistical analysis. And according to your suggestion, we have compared EDN1 as the control group to verify the changes in the expression levels of its downstream genes when the variables are wild-type EDNRB and mutant EDNRB. We hope that this analysis can support our conclusion.

3.    In figure2, I think there should be a control EDN1 only, for comparison to the two experimental groups.
Response: Thank you for your suggestions. We have changed the groups and set the group with only EDN1 as the control group, and re-analyzed it. The content of the manuscript has also been changed. Our specific modifications are: “The mean expression of the co-transfected wild-type EDNRB and EDN1 groups was about 8.17-fold than that of the EDN1 group. Raf and MITF expression levels were 1.39-fold and 4.46-fold higher (Figure 2E, 2F). Compared with the wild-type group, the expressions of PLCγ, Raf, and MITF in the mutant EDNRB group decreased by 2.04-fold, 2.19-fold and 1.79-fold, respectively (P-value < 0.01, t-test, Figure 2D, 2E, and 2F). The mean expression of PLCγ and MITF increased 4-fold and 2.48-fold than that of EDN1 group, respectively (Figure 2D, 2F). Interestingly, the expression of PLCγ in the co-transfection mutant EDNRB and EDN1 group was reduced by 1.58-fold compared with the EDN1 group. (Figure 2E). Therefore, it is inferred that mutant-type EDNRB will lead to the decrease of downstream gene expression, so it could affect the MITF gene that can regulate melanin expression” (lines 241-251).

4.Please add statistical analysis to the method section, or at least indicate the meaning of asterisks in the figure legends.
Response: Thank you for your valuable suggestions. We have indicated the meaning of **** in the legend as you requested. Our specific modifications are: “**** above the bars indicate significant differences at P-value < 0.01”.(lines 335, 344, 348-349)

5.There are many tiny issues, for example, “wild EDNRB” should be “wild-type EDNRB”.
Line 29-30: “expression of the three genes in mutant EDNRB cells” should be “expression of the three genes in cell line expressing mutant EDNRB”
In the discussion, mutant EDNRB “can’t bind to 395 EDN1, and then inhibit the expression of PLCγ, Raf, MITF, and other genes in the melanin 396 pathway”. In my understanding, loss of binding doesn’t necessarily lead to inhibit downstream gene expression. “failed to induce” would be more appropriate.
Response: We are very grateful for your professional comments on the manuscript. We have corrected the minor issues you raised. Your point of view is very correct. After thinking about it, we also feel that “failed to induce” is more appropriate, and we have changed it. Thank you for your comments.

6.Figure2 y axis “Elevate” should be “Relative”
Format issues, for example in 3.2, gene names, construct names and mRNA should be Italic.
Response: Thank you for reading our manuscript carefully. We have made changes to the questions you raised and corrected the format.

7.Figure3 and the according text, it’s the melanin getting measure rather than “melanin gene”, so it should be melanin production instead of melanin expression.
Response: We sincerely thank the reviewer for your careful reading. According to your suggestion, we have changed expression to production.

8.Line 212, tense “obtain”
Line 238, lowercase “while”
Response: We are very sorry for our careless mistake, thank you for your reminder, we have corrected it immediately.

9.Figure5A, lane miss labeling
Response: Thank you for your reminder. We are very sorry for our carelessness in forgetting this content. We have added it in time and hope to gain your understanding.